# Insights into the Corrosion Inhibition Performance of Plant Extracts of Different Genera in the Asteraceae Family for Q235 Steel in H_2_SO_4_ Medium

**DOI:** 10.3390/ijms26020561

**Published:** 2025-01-10

**Authors:** Tian-Shu Chu, Wen-Jie Mai, Hui-Zhen Li, Bo-Xin Wei, Yu-Qing Xu, Bo-Kai Liao

**Affiliations:** 1School of Chemistry and Chemical Engineering, Guangzhou University, Guangzhou 510006, China; tgd3rdroad368@e.gzhu.edu.cn (T.-S.C.); 32205170002@e.gzhu.edu.cn (W.-J.M.); lihz@gzhu.edu.cn (H.-Z.L.); 2Institute of Metal Research, Chinese Academy of Sciences, Shenyang 110016, China; bxwei17s@imr.ac.cn; 3School of Mechanical and Aerospace Engineering, Nanyang Technological University, Singapore 639798, Singapore; 4Joint Institute of Guangzhou University & Institute of Corrosion Science and Technology, Guangzhou University, Guangzhou 510006, China

**Keywords:** corrosion inhibitor, Asteraceae family, electrochemical method, Q235 steel

## Abstract

Nowadays, the development of plant extracts as corrosion inhibitors to protect metals from corrosion is a popular research direction. However, given the vast diversity of plant species in nature, it is imperative to explore effective methods to improve screening efficiency in order to quickly identify the corrosion inhibition potential of plants. In this work, a new strategy for developing plant-extracted eco-friendly corrosion inhibitors based on the family and genus of plants is proposed. Three plants of different genera in the Asteraceae family, including *Artemisia argyi* extract (AAE), *Chrysanthemum indicum* extract (CIE), and *Centipeda minima* extract (CME), were selected and successfully prepared as novel corrosion inhibitors for Q235 steel in a sulfuric acid solution. The corrosion inhibition behavior and corresponding mechanism were systematically investigated by using some electrochemical tests (open circuit potential, potentiodynamic polarization, and electrochemical impedance spectroscopy) and surface characterizations (Fourier transform infrared spectroscopy, scanning electron microscopy, and X-ray photoelectron spectroscopy). The experimental results illustrated that the main components of the three extracts were similar and that when combined with KI as mixed-type corrosion inhibitors, they could dramatically slow down the metal corrosion rate. The maximum value of the corrosion inhibition efficiency reached 96.29%, 96.50%, and 97.52%, respectively, at 200 mg/L and could increase to 98.64%, 97.65%, and 99.06%, respectively, with a prolonged immersion time. A synergistic effect exists between the three plant extracts and KI, leading to the firm adsorption of the three plant extract molecules onto a Q235 steel surface, thereby forming a robust protective film. This work demonstrated that plants of different genera in the Asteraceae family possessed similar corrosion inhibition capabilities, providing a novel way to select potential corrosion inhibitors from numerous plants based on family and genus classification.

## 1. Introduction

Carbon steel has been extensively utilized in diverse fields, such as construction engineering, mechanical equipment, oil and gas industries, and so on, on account of its exceptional mechanical properties and low cost [1,2,3]. In practical industry applications, the surface of carbon steel is prone to forming corrosion products and scale, which are usually cleaned and removed by pickling with hydrochloric acid, sulfuric acid, and nitric acid [4]. However, the pickling process inescapably corrodes the carbon steel [5,6,7]. Moreover, it is greatly crucial to inhibit the acid-driven corrosion of carbon steel during various service occasions [8].

Nowadays, diverse methods have been developed to protect carbon steel from corrosion, including anticorrosive coatings [9,10], sacrificial anode methods [11], cathodic protection methods [12], corrosion-resistant alloys [13], and corrosion inhibitor methods [14,15,16,17]. Among them, the corrosion inhibitor method has the advantages of a lower dosage, high efficiency, convenient operation, etc., and is recognized as one of the most prevalent adopted and valid methods for mitigating metal corrosion in acidic circumstances [18,19,20,21]. Nevertheless, the toxicity of many traditional inhibitors has been heeded, considering the environmental pollution and restrictions or bans. Therefore, it is essential to investigate non-toxic or less harmful substitutes to replace conventional detrimental inhibitors [16,22].

Owning to the superiorities of a wide distribution of raw materials, having a low cost, being eco-friendly, being degradable [23], and contributing no pollution [24], applications of plant-based extracts as corrosion inhibitors have become a prominent area of emphasis in scientific research. Examples include Eupatorium Adenophora Spreng [23], Modified nano-lignin [25], *Calotropis giganteen* [26], Feverfew root [27], *Acanthopanax senticosus* [28], Kapok [29], *Kleinhovia hospita* [30], etc. These studies attribute the inhibitory effects of plant extracts to their complex composition, which includes tannins, alkaloids, flavonoids, and glycosides [26]. The molecular structures of these compounds are characterized by a high density of heteroatoms (such as N, O, S, and P) and electronegative groups, serving as primary adsorption centers for the active components of plant extracts [23,25,29]. However, given the vast numbers of plants in nature, it would require a great amount of manpower, material resources, and financial investment to evaluate the corrosion inhibition performances of different plant extracts one by one through experimental testing. Therefore, it is imperative to explore efficient methods for enhancing screening efficiency.

Based on the principles of plant taxonomy, plants belonging to the same species or genus often share similar structures and compositions, which have a decisive impact on their corrosion inhibition performance and have positive guiding significance for preliminarily judging the potentiality of plants. Up to now, there are few summaries reporting on the similar corrosion inhibition performance of plant extracts from different genera within the same family, which has also rarely received attention. In light of this, according to the criterion of different genera within the same family, a substantial number of studies pertaining to plant extract inhibitors are investigated in this article, especially those from Lamiaceae [1,7,31,32,33,34,35], Leguminosae [8,36,37,38,39,40,41], Gramineae [42,43,44,45,46], Cucurbitaceae [9,47,48,49,50,51], Rosaceae [52,53,54,55,56], and Compositae [57,58,59,60,61,62,63,64,65]. It can be found that the components with inhibition properties in plant extracts from different genera within the same family are similar, and the overall corrosion inhibition efficiency is comparable. However, the relevance of the classification of plant genera in the same families and their corrosion inhibition ability is still unclear, considering the different testing conditions.

In this paper, three unexplored plant extracts of *Artemisia argyi*, *Chrysanthemum indicum*, and *Centipeda minima* from Asteraceae plants were selected to verify this observation in a sulfuric acid solution. In detail, a comprehensive investigation was conducted to examine the corrosion inhibition properties of the three plant extracts with or without KI on Q235 steel in 0.5 M H_2_SO_4_ solution. The corrosion inhibition performance and adsorption behavior of the three plant extracts were systematically investigated using several tests. Fourier transform infrared spectroscopy (FTIR) was used to analyze the structural information of the plant extracts. Detailed electrochemical analyses, including open circuit potential (OCP), potentiodynamic polarization (PDP), and electrochemical impedance spectroscopy (EIS), were performed to assess the protective ability. After exposure to a corrosive medium, scanning electron microscopy (SEM) and X-ray photoelectron spectroscopy (XPS) were employed to examine the morphology and composition of corrosion products on metal surfaces. Furthermore, the corrosion inhibition mechanism was deeply explored. This research is valuable for establishing a dependable pre-screening strategy to determine the potential corrosion inhibitors based on the family and genus classification of plants, thereby enhancing the efficiency of developing novel plant extract-based corrosion inhibitors.

## 2. Results and Discussion

### 2.1. Characterizations of AAE, CIE, and CME

The chemical groups of AAE, CIE, and CME were characterized using FT-IR. As shown in Figure 1, the spectra of AAE, CIE, and CME were quite similar, with most parts showing alike characteristic peaks. The broad peaks around 3385 and 3386 cm^−1^ could belong to the O-H stretching vibration of the glycoside molecules [66,67]. The obvious adsorption peaks near 2926, 2932, and 2922 cm^−1^ could be ascribed to the extra-ring substituent C-H stretching vibration [66,67]. The sharp peaks at 1650, 1644, and 1656 cm^−1^ could match the expansion and vibration peak of the carboxyl C=C double bond [66,68]. The double peaks near 1426, 1417, and 1383 cm^−1^ could be ascribed to C-H in-plane and O-H bending vibration, respectively. Peaks at 1284 and 1246 cm^−1^ were associated with the C-O-C stretching vibration of flavonoids [67,69]. Peaks around 1018, 1026, and 1024 cm^−1^ were related to C-O stretching perturbation in the phenolic groups [66,70]. The peaks near 573, 589, and 574 cm^−1^ were consistent with O-H out-of-plane bending vibrations on the aromatic rings. The majority of the structures of AAE, CIE, and CME exhibited similarities to those found in other members of the Asteraceae family [57,58,60,62]. These characteristic peaks indicated that all three extracts of AAE, CIE, and CME had the same main components of flavonoids, saponins, and polysaccharide organics, which proved that the main components of three different genera of composite plants were similar. The main components were also found in other Asteraceae family plants [61].

### 2.2. Open Circuit Potential (OCP) Curves

Figure 2 shows the changes in the OCP value with the immersion time for different concentrations of AAE, AAE + KI, CIE, CIE + KI, CME, and CME + KI. The OCP value exhibited minimal variation after the 1800s, indicating the attainment of a stable state at the interface between Q235 steel and its solution [27,71]. In addition, OCP showed negative displacement under various concentrations of the three extracts with or without KI, which manifested all the corrosion inhibitors mainly affected the cathodic reaction of Q235 steel and effectively retarded acid-induced corrosion. For example, with the addition of AAE under three different concentrations of 50, 100, and 200 mg/L, the stabilized OCP values exhibited a negative shift to −0.477, −0.480, and −0.487 V, respectively, in comparison with the blank sample of −0.457 V. The stabilized OCP values of the rest of the samples in this test also shifted negatively, ranging from 0.01 to 0.03 V.

### 2.3. Potentiodynamic Polarization (PDP) Curves

The PDP curves of Q235 steel are presented in Figure 3 under blank conditions and different concentrations of AAE, CIE, and CME (50, 100, and 200 mg/L) with or without the addition of 60 mg/L of KI (hereinafter referred to as compounding) at room temperature in 0.5 mol/L H_2_SO_4_ solution. Values of the corrosion potential (*E*_corr_), corrosion current density (*i*_corr_, mA·cm^−2^), anode Tafel slope (*β*_a_), cathode Tafel slope (*β*_c_), and *η*_PDP_ obtained from PDP curves. The *η*_PDP_ could be calculated from Equation (1) [17], as shown in Table 1.(1)$$\eta_{PDP}=\frac{i_{corr}^{0}-i_{corr}^{1}}{i_{corr}^{0}}\times 100\%$$
where *η* denotes the corrosion inhibition efficiency, $i_{corr}^{0}$ represents the corrosion current density in the blank solution, and *i*_corr_ indicates the corrosion current density when adding inhibitors.

As shown in Figure 3(a1,b1,c1) and Table 1, with the increase in inhibitor concentration, *E*_corr_ changed less than 0.02 V in the negative direction compared to that in the blank solution. *i*_corr_ partially decreased, accompanied by a low *η*_PDP_ value. The degree of anodic reaction inhibition was not significant, whereas the cathodic reaction exhibited minimal inhibition, which shows the three extracts mainly inhibited cathodic reactions. After the three extracts were combined with KI, as shown in Figure 3(a2,b2,c2), values of *E*_corr_ were nearly invariable with the rising concentration of the three extracts. However, both cathodic and anodic reactions were pronouncedly delayed. *i*_corr_ was obviously diminished, while *η*_PDP_ was substantially enhanced. The minimum *i*_corr_ of AAE + KI, CIE + KI, and CME + KI at the concentration of 200 mg/L were 0.0401, 0.0516, and 0.0210 mA·cm^−2^, singly, and corresponding maximal *η*_PDP_ were 96.69%, 95.74%, and 98.26%. According to the change range of *E*_corr_, the corrosion inhibitor could be divided into anodic, cathodic, and mixed types. The change values of *E*_corr_ in AAE + KI, CIE + KI, and CME + KI under different concentrations were less than 85 mV. Therefore, the three extracts and their compounds with KI were geared toward mixed-type corrosion inhibitors.

### 2.4. Electrochemical Impedance Spectroscopy (EIS) Curves

Figure 4 displays EIS results, containing Nyquist plots, Bode plots, and phase angle plots, with blank and three different concentrations of AAE, CIE, and CME with or without 60 mg/L of KI. The Nyquist plots in Figure 4(a1,c1,e1) exhibit large capacitive reactance arcs in high-frequency regions and small inductive reactance arcs in low-frequency regions. The capacitive reactance arcs were flat, resulting from the non-uniformity of the metal surface. As the concentration of the three extracts increased, the radius of the capacitive reactance arc and impedance mode at 0.01 Hz gradually increased while the phase angle range widened. However, there were no changes in the shape of Nyquist plots, suggesting that a protective film layer formed on the Q235 steel surface, hindering the corrosion process without altering the electrochemical corrosion mechanism. When the three extracts at various concentrations were combined with 60 mg/L of KI, all Nyquist plots comprised a large capacitive reactance arc in high-frequency regions and a small one in low-frequency regions without a minor inductive reactance arc, as shown in Figure 4(b1,d1,f1). When the concentration of three extracts increased, the shape of the Nyquist plots had no alteration observed, which indicated that the electrochemical corrosion mechanism of Q235 steel was not affected by the three compound corrosion inhibitors.

Meanwhile, both the capacitive reactance arc radius and impedance mode value at 0.01 Hz increased by degrees while the phase angle range broadened, suggesting an enhancement in the corrosion resistance properties of the three compound corrosion inhibitors. Compared to either the three extracts with different concentrations or 60 mg/L of KI solution alone, the three compound corrosion inhibitors displayed a significant increase in the capacitive reactance arc radius and phase angle range. Furthermore, the impedance mode increased by an order of magnitude, manifesting effective adsorption of the three compound corrosion inhibitors on the Q235 steel surface and the formation of a more secure protective layer. In order to better analyze EIS data, the equivalent circuits (a) and (b) depicted in Figure 5 were employed to accurately fit the curves (b1), (d1), and (f1) with KI, as well as the curves (a1), (c1), and (e1) without KI. The comprehensive fitting results are presented in Table 2. Among them, *R*_s_, *CPE*_dl_, *n,* and *R*_ct_ represent the solution resistance, double layer constant phase angle element, dispersion coefficient, and charge transfer resistance, respectively. *L* and *R*_L_ stand for inductance and inductive resistance, respectively. The value of n can be used to evaluate the deviation from the ideal behavior. When *n* was close to 1, it indicated capacitance. As *n* approached 0, it was roughly equal to a resistance. The corrosion inhibition efficiency *η*_EIS_ was calculated as shown in Equation (2) [72].(2)$$\eta_{EIS}=\frac{R_{ct}^{1}-R_{ct}^{0}}{R_{ct}^{1}}\times 100\%$$
where $R_{ct}^{0}$ and $R_{ct}^{1}$ are the blank transfer resistance and the transfer resistance after adding inhibitors, respectively.

As displayed in Table 2, the increase in inhibitor concentration led to a gradual decrease in *CPE*_dl_, a piecemeal increase in *R*_ct_, and an accompanying enhancement in *η*_EIS_. Particularly, the *CPE*_dl_ of three extracts at different concentrations with KI were smaller compared to their individual usage, while *R*_ct_ and *η*_EIS_ exhibited the inverse trends. These suggested the presence of a synergistic effect between the three extracts and KI.

When the concentration was 200 mg/L, the three compound inhibitors showed the highest inhibition efficiency *η*_EIS_, which were 96.29%, 96.50%, and 97.52%, respectively.

### 2.5. Long-Term Corrosion Inhibition Performance

Generally, it was of great importance to monitor their long-term corrosion inhibition performance as the corrosion inhibition ability of the corrosion inhibitor changed with the immersion time. The long-term corrosion inhibition behaviors of AAE, CIE, and CME under 200 mg/L with 60 mg/L of KI soaked in 0.5 mol/L H_2_SO_4_ solution for various times were evaluated using EIS tests, as shown in Figure 6. Evidently, all the Nyquist plots of three corrosion inhibitors with KI displayed one semicircular arc of capacitive reactance, suggesting a solitary time constant and corrosion process existing on the metal/electrolyte interface. The semicircular arc appeared flat, which was typically attributed to the frequency dispersion of interface impedance caused by surface roughness, impurities, inhibitor adsorption, and surface inhomogeneity. The capacitive reactance arc radius and the impedance modulus at 0.01 Hz increased initially, followed by a subsequent decrease. The largest values were observed at 4 h, which were significantly larger than those in blank and KI solutions. The overall Bode plots also exhibited the single-time-constant characteristic commonly associated with double-layer capacitors. It suggested that a denser protective layer formed by the three corrosion inhibitors hindered the penetration of corrosive media, resulting in an impedance spectrum that only reflected interface capacitance corresponding to charge transfer processes.

The electrochemical parameters of three corrosion inhibitors within 48 h are presented in Table 3. It was evident that the addition of the three compound corrosion inhibitors led to a rapid decrease in *CPE*_dl_. This could be attributed to the replacement of water molecules initially adsorbed on the Q235 steel surface by inhibitor molecules, which resulted in a gradual transformation from an adsorbed hydration layer to the corrosion inhibitor adsorption layer at the metal/solution interface. Due to the significantly higher dielectric constant and visibly thinner adsorption layer thickness of water molecules compared to those of inhibitors, the interface capacitive covered by the three compound inhibitor molecules was less than that of water molecules. Furthermore, when adding inhibitors, *R*_ct_ increased significantly and showed the same trend, indicating their excellent corrosion inhibition performance. Additionally, all three inhibitors exhibited similar trends in terms of their corrosion inhibition efficiency, whose maximum values after 4 h reached 98.64%, 97.65%, and 99.06%, respectively. Moreover, over a period spanning 48 h, the corrosion inhibition efficiencies of the three compound inhibitors maintained a relatively higher level above 89%, demonstrating their excellent long-term corrosion inhibition performances.

### 2.6. XPS Results

In order to further study the compositions of corrosion products in the presence of the three corrosion inhibitors, XPS experiments were carried out. The corrosion products were predominantly composed of C, O, Fe, and N elements, indicating successful adsorptions for the three corrosion inhibitor molecules on the Q235 steel surface. In Figure 7, the peak positions for the three corrosion inhibitors of AAE (left column), CIE (middle column), and CME (right column) from left to right are similar. The specific positions detailed as follows: the C1s spectra manifested three distinct peaks in Figure 7(a1,b1,c1), corresponding to C-C/C-H at 284.80, 284.80, and 284.90 eV; C-N/C-O at 285.95, 285.27, and 285.95 eV; and C=O at 285.30, 286.00, and 284.50 eV [73,74]. Fe2p spectra included two peaks in Figure 7(a2,b2,c2), Fe (II) at 710.76, 710.70, and 710.73 eV and Fe (III) at 712.65, 712.10, and 712.05 eV [75,76]. These two peaks could be ascribed to Fe_2_O_3_ and FeOOH, respectively. Figure 7(a3,b3,c3) displays the O1s spectra, containing three peaks; the first peaks were located near 529.58, 521.44, and 529.81 eV. These could be attributed to Fe-O, presumably related to the oxygen atom bound to iron oxide [25]. The second peak was at 531.00, 530.26, and 531.30 eV, which was inferred to be O-H [77,78]. The last peak was around 531.67, 532.44, and 531.89 eV, which was related to C=O [79]. Figure 7(a4,b4,c4) shows the N1s peaks, where one group’s peaks were located at 399.80, 400.18, and 399.95 eV [77,80], imputing to C-N of the active group in the three corrosion inhibitors. Another group peaks at 399.30, 399.60, and 399.40 eV, belonging to N-H. The XPS results proved the effective adsorption of the AAE, CIE, and CME molecules on the Q235 steel surface. The composition of corrosion products and the bonding mechanism can also be observed in other plants of the Asteraceae family [63].

### 2.7. SEM Observations

Figure 8 shows the microscopic morphologies of corroded Q235 steel after soaking in 0.5 mol/L of H_2_SO_4_ for 4 h under various conditions: blank, 60 mg/L of KI, and the three inhibitors of AAE, CIE, and CME at 200 mg/L with or without 60 mg/L of KI. Under blank conditions in Figure 8a, serious corrosion could be observed on the Q235 steel surface, which was characterized by an irregular and uneven form. The sole addition of 60 mg/L of KI, AAE, CIE, and CME at 200 mg/L did not significantly retard the corrosion of Q235 steel, and there were still numerous pits and severe corrosion present in Figure 8b–e. This observation suggested that the inhibition efficiency of 60 mg/L of KI and 200 mg/L of the three different extracts for Q235 steel remained relatively low, which aligned with the electrochemical results.

However, under the three compound solutions of 200 mg/L of three inhibitors with 60 mg/L in Figure 8f–h, a significant reduction in surface pits on the Q235 steel surface was observed, accompanied by the formation of an evident protective film. This further substantiated the relatively better corrosion inhibition ability of the three compound inhibitors for Q235 steel in sulfuric acid solution.

### 2.8. Corrosion Inhibition Mechanism of AAE, CIE, and CME

Figure 9 provides an illustrative depiction of the inhibition mechanism of AAE/CIE/CME on the Q235 steel surface in H_2_SO_4_ solution with or without KI. As for Q235 steel in sulfuric acid, the main cathodic reaction (hydrogen evolution reaction) and anodic reaction (anode dissolution) for electrochemical corrosion are as follows:Cathodic reaction: 2H^+^ + 2e^−^ → H_2_ ↑(3)Anodic reaction: Fe → Fe^2+^ + 2e^−^(4)

Owing to the abundant O or N-containing polar groups in the three plant-extracted inhibitors [15], protonation reactions occurred in sulfuric acid solution, as shown in Equations (5)~(7):AAE + xH^+^ → AAEH^x+^
(5)CIE + xH^+^ → CIEH^x+^(6)CME + xH^+^ → CMEH^x+^(7)

The protonation reaction could significantly reduce the reduction rate of H^+^, thereby decelerating the cathodic reaction [81]. The polar groups in AAE/CIE/CME and AAEH^x+^/CIEH^x+^/CMEH^x+^ coordinated with the Fe atoms on the surface of Q235 steel, allowing them to implement chemisorption onto Q235 steel surface. Additionally, the specific adsorption of SO_4_^2−^ on the Q235 steel surface resulted in its surface with a negative charge [82,83], which facilitated the adsorption of AAEH^x+^/CIEH^x+^/CMEH^x+^ by the electrostatic action. Upon adsorption, the AAEH^x+^/CIEH^x+^/CMEH^x+^ displaced water molecules and SO_4_^2−^, contributing to the reduction in the contact area between water molecules/sulfates and metal and thus mitigating the anodic dissolution reaction. However, considering the relatively lower corrosion inhibition efficiency, the adsorption films formed by AAEH^x+^/CIEH^x+^/CMEH^x+^ were relatively sparse and could not fully cover the entire surface of Q235 steel.

When KI was added, I^−^ may become oxidized by O_2_ under acidic conditions to form triiodide (I_3_^−^):6I^−^ + 4H + O_2_ → 2I_3_^−^ + 2H_2_O(8)

Because of the large size and easy polarization of I^−^ and I_3_^−^, they easily adhered to the Q235 steel surface, thereby forming a uniform layer of negative charge [84,85]. The negative charge layer replaced a portion of the initially adsorbed water molecules and SO_4_^2−^ on the Q235 steel surface, thus retarding the corrosion process of Q235 steel to some extent. When the three plant extracts were combined with KI, the corrosion inhibition efficiencies were significantly enhanced according to EIS and PDP results.

After adding KI, firstly, the anions of I^−^ and I_3_^−^ exhibited stronger specificity compared to SO_4_^2−^, leading to increased adsorption on the Q235 steel surface and subsequent augmentation of negative charge. Then, substantial adsorption of AAEH^x+^/CIEH^x+^/CMEH^x+^ onto the anionic layer formed by I^−^ and I_3_^−^ acting as a bridge. This resulted in the formation of a more comprehensive adsorption film layer that further amplified the corrosion inhibition effect. Additionally, the defects in the adsorption film could be filled by AAE/CIE/CME through chemisorption, resulting in a denser adsorption film and an enhanced corrosion inhibition effect.

## 3. Experiment

### 3.1. Material Preparations

The leaves of *Artemisia argyi*, *Chrysanthemum indicum*, and *Centipeda minima* were cut up slightly and wrapped with qualitative test paper. Then, each type of clipped leaf was immersed in an extraction solution with deionized water/anhydrous ethanol at a volume ratio of 1:1 several times. Thereafter, the three extracted solutions were subjected to Soxhlet extraction at 80 °C for 2 h [86,87]. The obtained extraction solutions were filtered and then dried in the oven at 100 °C to obtain the powder products of AAE, CIE, and CME.

The Q235 steel sheets were cut into size of 10 × 10 × 5 mm. The carbon steel sheets were connected and soldered with copper wire on one side, leaving an exposed area of 1 cm^2^ as rest surface sealed with epoxy resin. Prior to each test, the exposed area was polished using 220, 400, 800, 1200, and 2000 mesh sandpaper in sequence and cleaned with deionized water and acetone, followed by drying using cold air.

### 3.2. Electrochemical Tests

A CS350 electrochemical workstation (310H, Wuhan Corrtest Instrument Corp., Ltd. Wuhan, China) was used to perform the electrochemical experiments with a three-electrode system. Further, 35 steel functioned as the working electrode, saturated calomel electrode (SCE) with saturated potassium chloride solution served as the reference electrode, and Pt electrode with 4 cm^2^ acted as the counter electrode. Before each electrochemical test, an open circuit potential (OCP) test was executed at the end of reaching a steady state. The electrochemical impedance spectroscopy (EIS) test was conducted within the frequency range of 0.01 Hz to 10,000 Hz, employing an AC amplitude of 10 mV. The potentiodynamic polarization (PDP) test was examined at a scan rate of 0.5 mV/s, encompassing a potential range of ±150 mV relative to OCP.

### 3.3. Surface Characterization

The chemical constituents of AAE, CIE, and CME were analyzed using Fourier transform infrared spectroscopy (FT-IR, Spectrum 100, Nicolet 6700, Waltham, MA, USA). Scanning Electron Microscopy (JSM-7001F, JEOL, Tokyo, Japan) was employed to examine the microstructure of the corrosion products. X-ray photoelectron spectroscopy (XPS, Axis Nova, London, UK) was utilized to represent the composition of corrosion products.

## 4. Conclusions

In this study, three plant-extracted corrosion inhibitors of different genera in the Asteraceae family for Q235 steel in H_2_SO_4_ solution were prepared based on the family and genus of the plants. The corrosion inhibition performance and synergistic inhibitory effect of AAE, CIE, and CME with KI were investigated via some electrochemical measurements and surface characterizations. This work provided a novel and feasible strategy to efficiently develop plant-extracted eco-friendly corrosion inhibitors among numerous plants. The main conclusions were as follows:
(1)The characteristic peaks indicated that the primary constituents of flavonoids, saponins, and polysaccharide organic compounds in the three extracts of AAE, CIE, and CME were identical, which confirmed the similarity of the main components among the three extracts from different genera of composite plants.(2)The presence of AAE, CIE, CME, or KI alone mainly acted as a cathodic inhibitor, while the combination of the three extracts with KI as a mixed inhibitor further enhanced the inhibition of the cathodic reaction and anodic reaction. Additionally, when AAE, CIE, and CME were combined with KI, the capacitive reactance arc radius, impedance mode, and phase angle range exhibited significantly greater values compared to their individual usage.(3)AAE, CIE, CME, and KI exhibited moderate inhibitory effects on Q235 steel in 0.5 M of H_2_SO_4_. The combination of AAE, CIE, and CME with KI demonstrated a significant synergistic effect, resulting in inhibition efficiencies up to 96.29%, 96.50%, and 97.52%, respectively. Furthermore, the combination of the three extracts with KI displayed excellent long-term sustained release properties by maintaining high levels for 48 h, achieving the highest corrosion inhibition efficiency after 4 h of 98.64%, 97.65%, and 99.06%, respectively.(4)The XPS and SEM images confirmed the effective adsorption of AAE, CIE, and CME on the Q235 steel surface, resulting in the formation of a protective film. The combined synergistic effect of AAE, CIE, and CME with KI significantly retarded the corrosion process on the Q235 steel surface.


## Figures and Tables

**Figure 1 ijms-26-00561-f001:**
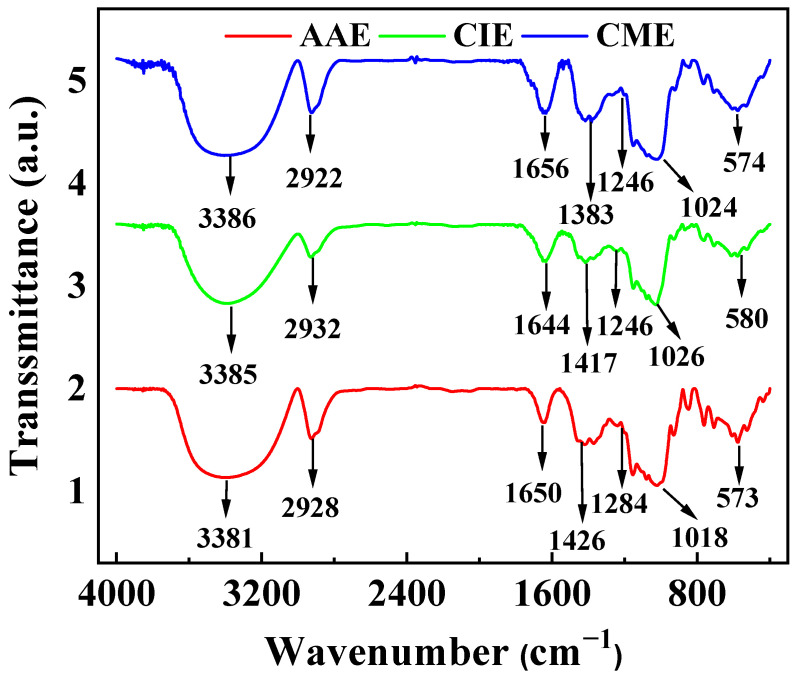
The FT-IR spectra of AAE, CIE, and CME.

**Figure 2 ijms-26-00561-f002:**
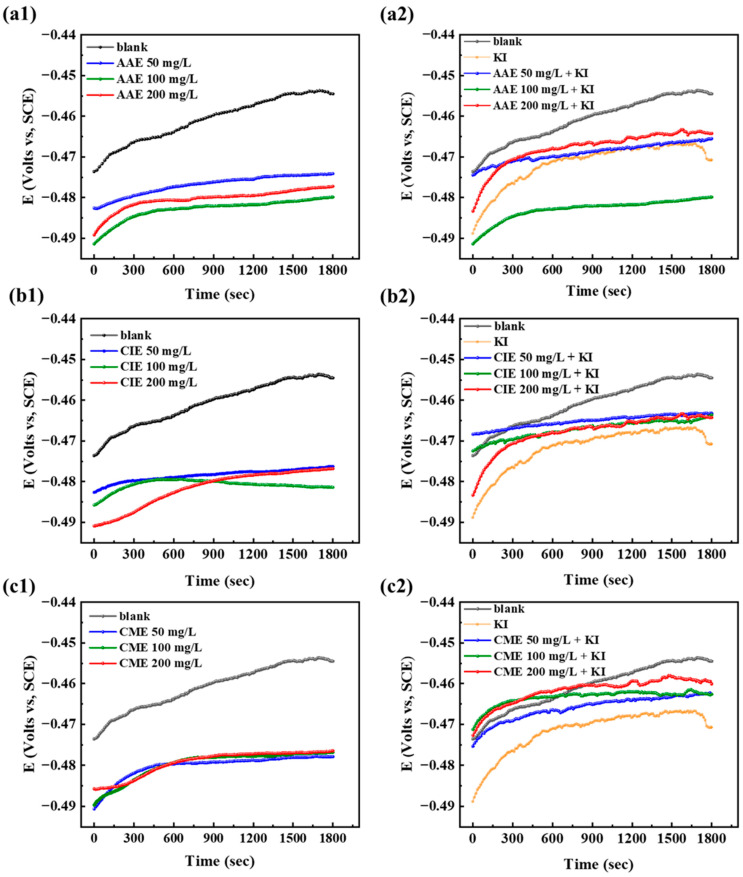
OCP of Q235 steel under different concentrations of AAE, CIE, and CME with or without KI in 0.5 mol/L H_2_SO_4_. (**a1**) Blank and three different concentrations of AAE without KI; (**a2**) blank, KI, and three different concentrations of AAE with KI; (**b1**) blank and three different concentrations of CIE without KI; (**b2**) blank, KI, and three different concentrations of CIE with KI; (**c1**) blank and three different concentrations of CME without KI; (**c2**) blank, KI, and three different concentrations of CME with KI.

**Figure 3 ijms-26-00561-f003:**
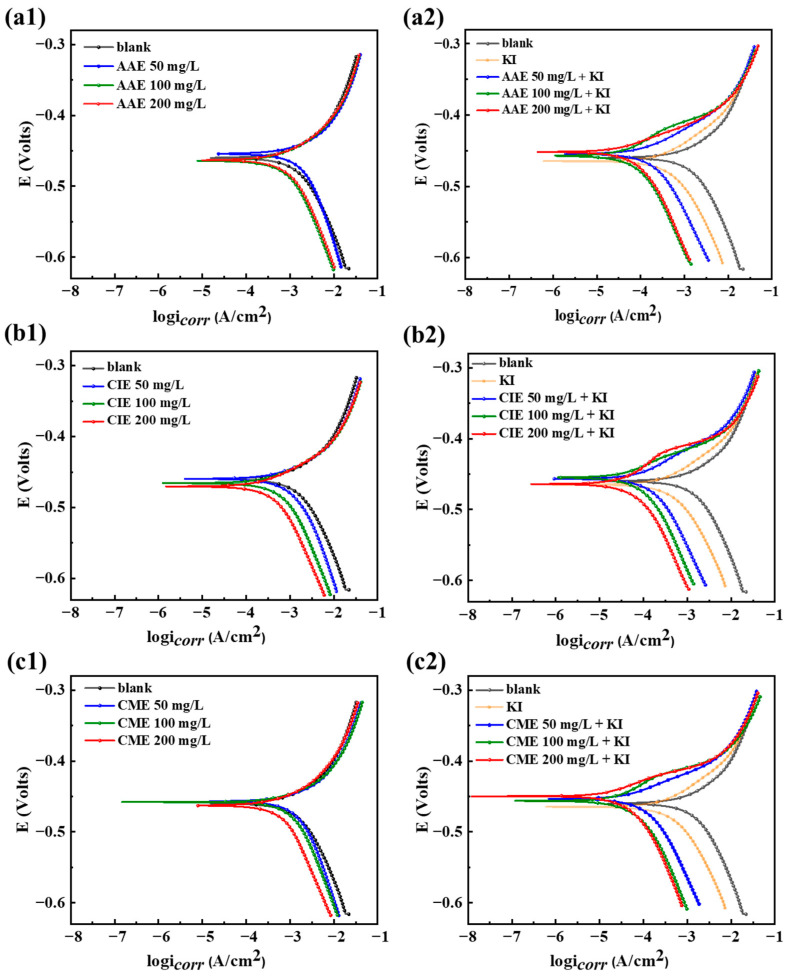
PDP curves of Q235 steel in 0.5 mol/L H_2_SO_4_ under different concentrations of AAE, CIE, and CME with and without KI; (**a1**) blank and three different concentrations of AAE without KI; (**a2**) blank, KI, and three different concentrations of AAE with KI; (**b1**) blank and three different concentrations of CIE without KI; (**b2**) blank, KI, and three different concentrations of CIE with KI; (**c1**) blank and three different concentrations of CME without KI; (**c2**) blank, KI, and three different concentrations of CME with KI.

**Figure 4 ijms-26-00561-f004:**
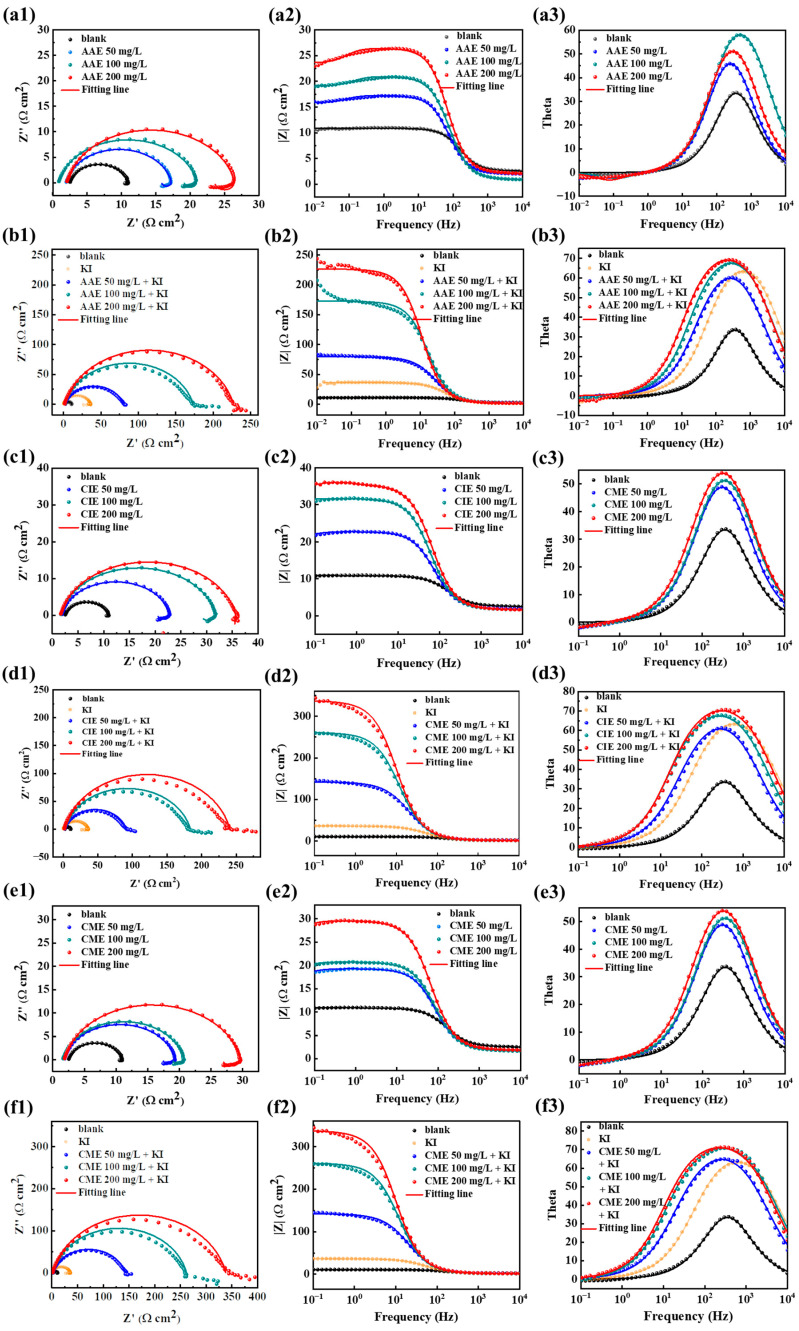
EIS curves of Q235 steel in 0.5 mol/L H_2_SO_4_ under different concentrations of AAE, CIE, and CME with and without KI; (**a1**–**a3**) Nyquist plots, Bode plots, and phase angle plots for blank, and three different concentrations of AAE without KI; (**b1**–**b3**) Nyquist plots, Bode plots and phase angle plots for blank, KI, and three different concentrations of AAE with KI; (**c1**–**c3**) Nyquist plots, Bode plots, and phase angle plots for blank and three different concentrations of CIE without KI; (**d1**–**d3**) Nyquist plots, Bode plots, and phase angle plots for blank, KI, and three different concentrations of CIE with KI; (**e1**–**e3**) Nyquist plots, Bode plots, and phase angle plots for blank and three different concentrations of CME without KI; (**f1**–**f3**) Nyquist plots, Bode plots, and phase angle plots for blank, KI, and three different concentrations of CME with KI.

**Figure 5 ijms-26-00561-f005:**
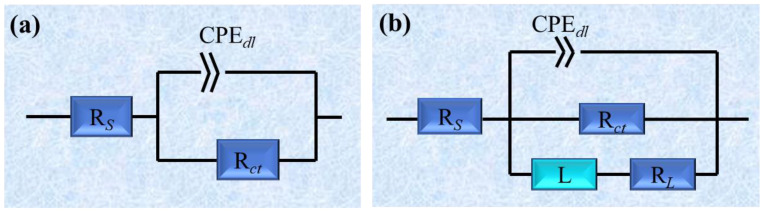
Equivalent circuit diagram of EIS curves of Q235 steel in 0.5 mol/L H_2_SO_4_ under different extracts with different extract concentrations: (**a**) blank, KI, and the compounded inhibitor with KI; (**b**) the uncompounded inhibitor without KI.

**Figure 6 ijms-26-00561-f006:**
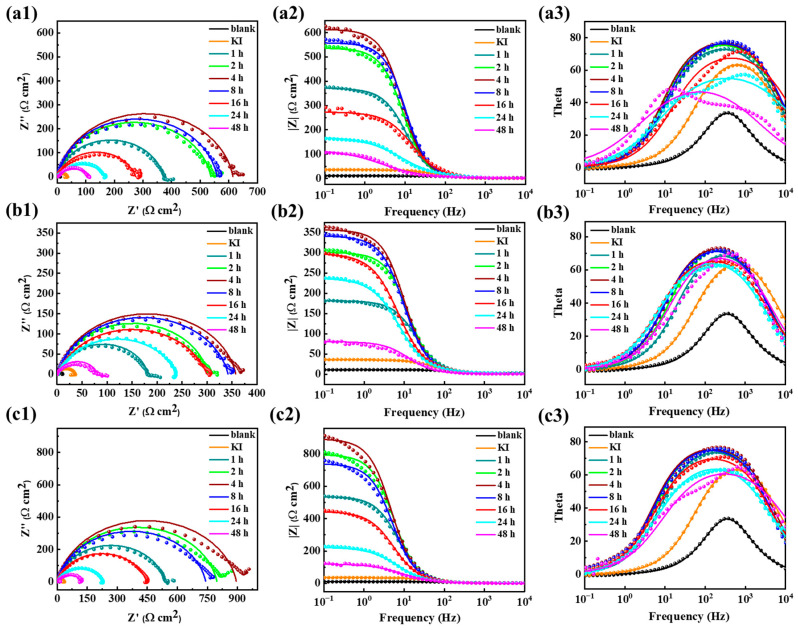
EIS curves of Q235 steel in 0.5 mol/L H_2_SO_4_ at 200 mg/L AAE, CIE, and CME combined with 60 mg/L of KI under different immersion times; (**a1**–**a3**) Nyquist plots, Bode plots, and phase angle plots for blank, KI, and 200 mg/L AAE with 60 mg/L of KI under different immersion times; (**b1**–**b3**) Nyquist plots, Bode plots, and phase angle plots for blank, KI, and 200 mg/L of CIE with 60 mg/L of KI under different immersion times; (**c1**–**c3**) Nyquist plots, Bode plots, and phase angle plots for blank, KI, and 200 mg/L of CME with 60 mg/L of KI under different immersion times.

**Figure 7 ijms-26-00561-f007:**
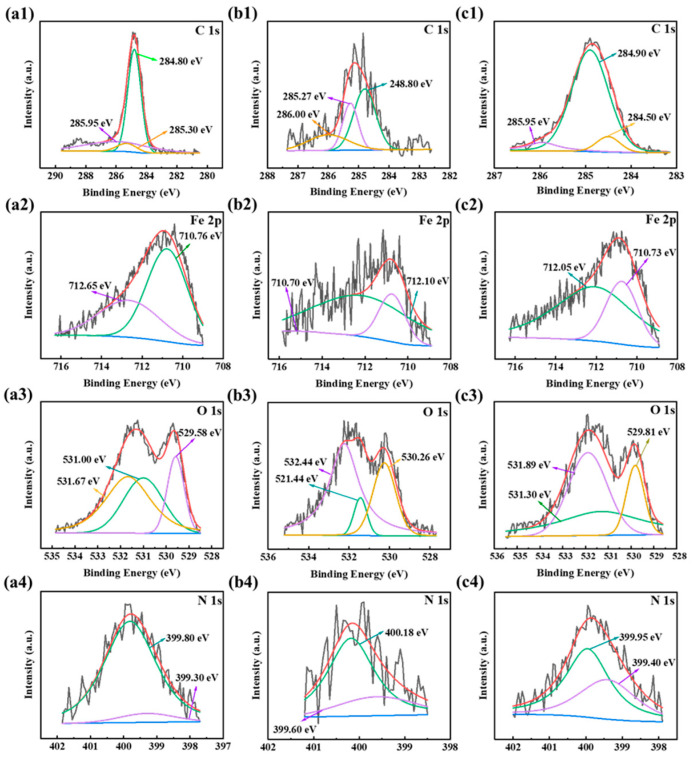
XPS image of corrosion products soaked in AAE, CIE, and CME under 200 mg/L combined with 60 mg/L of KI; (**a1**–**a4**) C1s, Fe2p, O1s, and N1s for 200 mg/L of AAE with 60 mg/L of KI; (**b1**–**b4**) C1s, Fe2p, O1s, and N1s for 200 mg/L of CIE with 60 mg/L of KI; (**c1**–**c4**) C1s, Fe2p, O1s, and N1s for 200 mg/L of CME with 60 mg/L of KI.

**Figure 8 ijms-26-00561-f008:**
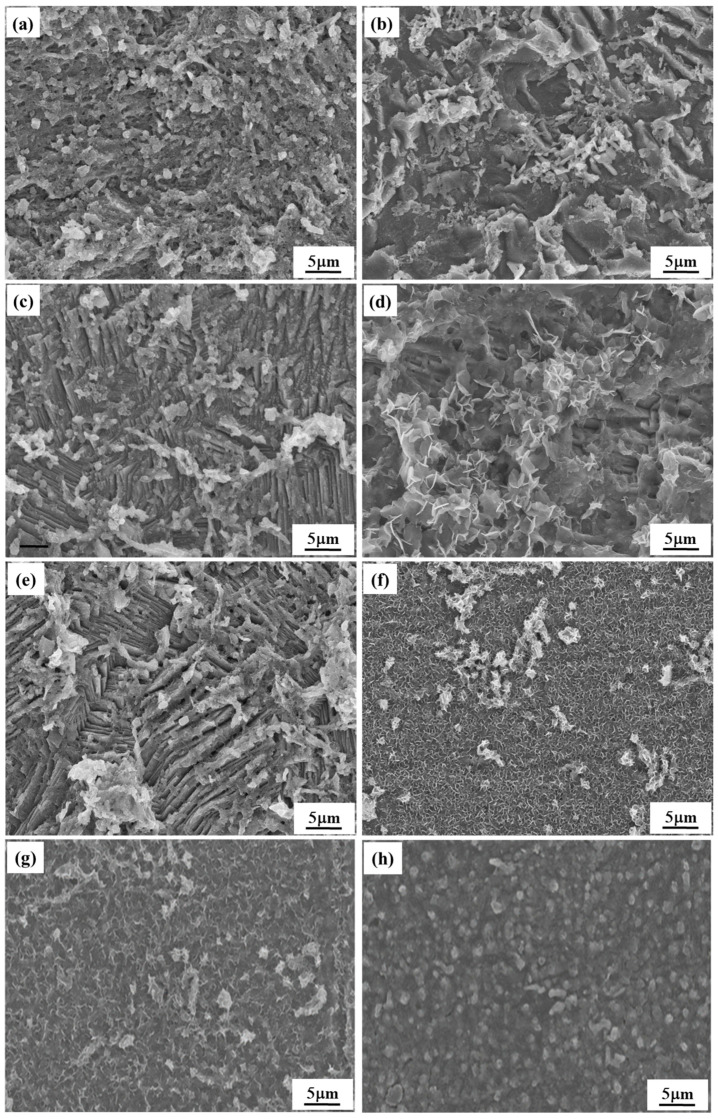
SEM images of corroded Q235 steel surface after soaking in 0.5 mol/L of H_2_SO_4_ for 4 h under the conditions of blank, 60 mg/L of KI, and 200 mg/L different extracts: (**a**) blank; (**b**) 60 mg/L of KI; (**c**) 200 mg/L of AAE; (**d**) 200 mg/L of CIE; (**e**) 200 mg/L of CME; (**f**) 200 mg/L of AAE with 60 mg/L of KI; (**g**) 200 mg/L of CIE with 60 mg/L of KI; (**h**) 200 mg/L of CME with 60 mg/L of KI.

**Figure 9 ijms-26-00561-f009:**
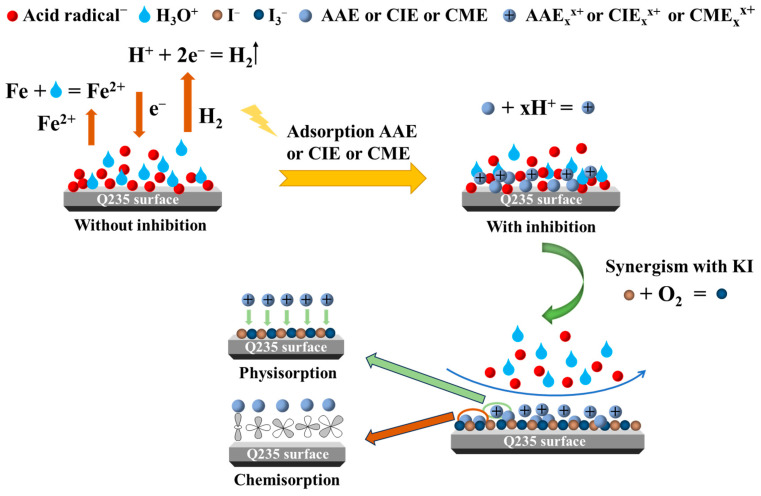
Diagram of the corrosion inhibition mechanism of AAE, CIE, and CME with KI.

**Table 1 ijms-26-00561-t001:** Electrochemical parameters of Q235 steel in 0.5 mol/L H_2_SO_4_ with different extracts at various concentrations.

	Concentration (mg/L)	*E*_corr_ (V)	*i*_corr_ (mA·cm^−2^)	*η* (%)
blank	--	−0.46	1.21 ± 0.06	--
KI	--	−0.47	0.32 ± 0.009	73.30 ± 0.15
AAE	50	−0.45	0.782 ± 0.025	35.87 ± 0.59
	100	−0.46	0.63 ± 0.032	47.77 ± 0.10
	200	−0.46	0.67 ± 0.027	44.63 ± 0.25
AAE + 60 mg/L of KI	50	−0.45	0.12 ± 0.0054	90.25 ± 0.21
	100	−0.46	0.08 ± 0.0013	93.69 ± 0.32
	200	−0.45	0.04 ± 0.0025	96.69 ± 0.01
CIE	50	−0.46	0.64 ± 0.03	47.02 ± 0.17
	100	−0.47	0.44 ± 0.011	63.55 ± 0.05
	200	−0.47	0.32± 0.0076	73.55 ± 0.09
CIE + 60 mg/L of KI	50	−0.46	0.15 ± 0.0016	87.93 ± 0.36
	100	−0.45	0.07 ± 0.001	94.36 ± 0.16
	200	−0.46	0.05 ± 0.0008	95.74 ± 0.11
CME	50	−0.46	1.11 ± 0.0016	8.26 ± 0.23
	100	−0.46	0.69 ± 0.002	43.22 ± 0.39
	200	−0.46	0.49 ± 0.01	59.42 ± 0.06
CME + 60 mg/L of KI	50	−0.45	0.06 ± 0.0002	94.97 ± 0.06
	100	−0.46	0.05 ± 0.0045	95.67 ± 0.17
	200	−0.45	0.02 ± 0.006	98.26 ± 0.06

**Table 2 ijms-26-00561-t002:** Fitting results and corrosion inhibition efficiency of impedance spectra at different concentrations of different extracts.

	Concentration (mg/L)	*R*_S_ (Ω cm^2^)	*CPE*_dl_ (μF cm^−2^)	*n*	*R*_L_ (Ω cm^2^)	*L* (µH cm^2^)	*R*_ct_ (Ω cm^2^)	*η*_EIS_ (%)
blank	--	2.53 ± 0.27	2.31 × 10^−4^	0.90	--	--	8.36 ± 0.43	--
KI	--	0.60 ± 0.12	1.81 × 10^−4^	0.86	--	--	35.27 ± 6.84	76.30 ± 1.17
AAE	50	1.98 ± 0.2	2.54 × 10^−4^	0.90	152.62 ± 13.54	319.91	15.30 ± 2.17	45.36 ± 1.53
	100	0.85 ± 0.03	1.82 × 10^−4^	0.89	200.21 ± 10.81	257.16	20.16 ± 2.62	58.53 ± 0.99
	200	2.06 ± 0.22	1.75 × 10^−4^	0.90	184.86 ± 25.82	400.03	24.42 ± 2.89	65.77 ± 0.70
AAE + KI 60 mg/L	50	1.82 ± 0.14	1.68 × 10^−4^	0.84	--	--	77.84 ± 8.92	89.26 ± 0.21
	100	1.48 ± 0.08	1.19 × 10^−4^	0.86	--	--	171.53 ± 30.18	95.13 ± 0.20
	200	1.37 ± 0.15	9.14 × 10^−5^	0.86	--	--	225.60 ± 48.34	96.29 ± 0.22
CIE	50	1.85 ± 0.26	1.75 × 10^−4^	0.91	171.45 ± 19.25	490.80	20.96 ± 2.87	60.11 ± 1.04
	100	1.63 ± 0.07	1.54 × 10^−4^	0.90	295.80 ± 35.11	2192.00	30.08 ± 5.68	72.21 ± 1.31
	200	1.60 ± 0.33	1.33 × 10^−4^	0.90	133.96 ± 21.08	1419.36	34.06 ± 6.33	75.46 ± 1.13
CIE + KI 60 mg/L	50	2.02 ± 0.11	1.31 × 10^−4^	0.85	--	--	88.93 ± 12.09	90.60 ± 0.24
	100	1.58 ± 0.18	1.00 × 10^−4^	0.86	--	--	182.46 ± 34.23	95.41 ± 0.20
	200	1.51 ± 0.13	9.97 × 10^−5^	0.88	--	--	239.50 ± 63.71	96.50 ± 0.31
CME	50	1.86 ± 0.41	1.93 × 10^−4^	0.91	144.62 ± 10.12	399.31	17.46 ± 2.26	52.12 ± 1.14
	100	1.62 ± 0.24	1.87 × 10^−4^	0.90	184.34 ± 19.8	588.83	19.14 ± 2.88	56.32 ± 1.36
	200	1.83 ± 0.31	1.63 × 10^−4^	0.89	252.10 ± 26.4	885.06	27.82 ± 4.11	69.95 ± 0.90
CME + KI 60 mg/L	50	1.83 ± 0.16	1.13 × 10^−4^	0.86	--	--	140.72 ± 29.63	94.06 ± 0.34
	100	1.25 ± 0.06	8.87 × 10^−5^	0.87	--	--	260.44 ± 45.15	96.79 ± 0.13
	200	1.54 ± 0.13	7.80 × 10^−5^	0.87	--	--	336.54 ± 58.48	97.52 ± 0.10

**Table 3 ijms-26-00561-t003:** Fitting results of EIS curves of Q235 steel soaking time in 0.5 mol/L H_2_SO_4_ at 200 mg/L combined with 60 mg/L of KI at different immersion times.

	Time (h)	*R*_s_ (Ω cm^2^)	*CPE*_dl_ (µF cm^−2^)	*n*	*R*_ct_ (Ω cm^2^)	*η*_EIS_ (%)
blank	--	2.53 ± 0.27	2.31 × 10^−4^	0.90	8.36 ± 0.43	--
KI	--	0.60 ± 0.12	1.81 × 10^−4^	0.86	35.27 ± 6.84	76.30 ± 1.17
AAE 200 mg/L + KI 60 mg/L	1	1.25 ± 0.11	6.64 × 10^−5^	0.88	370.83 ± 58.54	97.75 ± 0.08
	2	1.25 ± 0.15	5.03 × 10^−5^	0.90	536.12 ± 84.98	98.44 ± 0.05
	4	1.32 ± 0.09	4.36 × 10^−5^	0.90	613.14 ± 91.51	98.64 ± 0.05
	8	1.31 ± 0.13	4.63 × 10^−5^	0.91	557.76 ± 63.40	98.50 ± 0.03
	16	1.03 ± 0.04	8.63 × 10^−5^	0.82	273.33 ± 40.26	96.94 ± 0.09
	24	0.92 ± 0.05	1.19 × 10^−4^	0.69	169.71 ± 20.04	95.07 ± 0.10
	48	1.70 ± 0.21	1.44 × 10^−4^	0.64	116.90 ± 10.80	92.85 ± 0.10
CIE 200 mg/L + KI 60 mg/L	1	1.83 ± 0.26	1.05 × 10^−4^	0.88	180.26 ± 26.88	95.36 ± 0.14
	2	1.87 ± 0.20	6.92 × 10^−5^	0.89	299.93 ± 58.92	97.21 ± 0.14
	4	1.87 ± 0.17	6.75 × 10^−5^	0.89	355.70 ± 65.69	97.65 ± 0.11
	8	1.86 ± 0.24	6.88 × 10^−5^	0.88	340.39 ± 79.61	97.54 ± 0.17
	16	1.80 ± 0.20	7.84 × 10^−5^	0.82	297.71 ± 60.08	97.19 ± 0.15
	24	1.79 ± 0.18	1.10 × 10^−4^	0.81	236.50 ± 52.80	96.47 ± 0.23
	48	0.57 ± 0.05	1.62 × 10^−4^	0.84	80.27 ± 16.26	89.59 ± 0.56
CME 200 mg/L + KI 60 mg/L	1	1.89 ± 0.22	6.32 × 10^−5^	0.89	535.76 ± 101.03	98.44 ± 0.07
	2	1.85 ± 0.24	3.97 × 10^−5^	0.90	796.04 ± 104.89	98.95 ± 0.03
	4	1.84 ± 0.13	3.27 × 10^−5^	0.90	893.13 ± 113.25	99.06 ± 0.02
	8	1.85 ± 0.17	4.11 × 10^−5^	0.89	739.92 ± 108.40	98.87 ± 0.03
	16	1.56 ± 0.19	6.62 × 10^−5^	0.84	444.41 ± 80.25	98.12 ± 0.08
	24	1.28 ± 0.10	1.12 × 10^−4^	0.79	223.46 ± 42.26	96.26 ± 0.18
	48	0.59 ± 0.05	1.24 × 10^−4^	0.75	120.60 ± 15.84	93.07 ± 0.17

## Data Availability

Data are contained within the article.

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
