# Peer review of "Insights into the Corrosion Inhibition Performance of Plant Extracts of Different Genera in the Asteraceae Family for Q235 Steel in H2SO4 Medium"

_ijms, 2025, doi:10.3390/ijms26020561_

Round 1
Reviewer 1 Report
Comments and Suggestions for Authors
1.What are the main components of the three plant extracts (AAE, CIE, and CME) and how do they contribute to the corrosion inhibition process?
2.How does the combination of the three plant extracts with KI enhance the corrosion inhibition efficiency compared to their individual use?
3.What is the significance of the observed synergistic effect between the plant extracts and KI in terms of the adsorption behavior on the Q235 steel surface?
4.How do the electrochemical impedance spectroscopy (EIS) results support the formation of a protective layer by the corrosion inhibitors and what information can be obtained from the equivalent circuits used for data fitting?
5.What are the possible reasons for the long-term corrosion inhibition performance of the three compound inhibitors and how does it relate to the stability of the adsorption film?
6.Based on the results of this study, can you propose other potential applications or areas of research for plant-extracted corrosion inhibitors from the Asteraceae family?
Author Response
Comments 1: What are the main components of the three plant extracts (AAE, CIE, and CME) and how do they contribute to the corrosion inhibition process?
Response 1: Thank you for pointing this out. We agree with this comment. The main components of the three plant extracts (AAE, CIE and CME) are flavonoids, saponins and polysaccharides based on the “Characterizations of AAE, CIE and CME” results in section 3.1 from the line of 133 to 150 of the revised manuscript.
The flavonoids, saponins and polysaccharides contain the heteroatoms such as N, O, S, P, which could act as adsorption sites for the corrosion inhibitor molecules to adsorb onto the active sites of the metal, thus contributing to the corrosion inhibition process. These conclusions have been confirmed by the published literature of [23, 25, 26, 29] in the Introduction part from the line of 61 to 65 of the revised manuscript. The corresponding expression has been added in the Introduction part. In addition, these conclusions are further explained in the corresponding sections of 3.6 and 3.8 of the revised manuscript.
The added details of the flavonoids, saponins and polysaccharides contributing to the corrosion inhibition process in Introduction part as follows:
“These studies attribute the inhibitory effects of plant extracts to their complex composition, which includes tannins, alkaloids, flavonoids, and glycosides [26]. The molecular structures of these compounds are characterized by a high density of heteroatoms (such as N, O, S, P) and electronegative groups, serving as primary adsorption centers for the active components of plant extracts [23, 25, 29].”
[23] Zhou, Y.; Zhu, C.; Xu, S.Y.; Xiang, B.; Marzouki, R. Combining electrochemical, surface topography analysis, and theoretical alculation methods to insight into the anti-corrosion property of Syzygium samarangense leaf extract. J. Ind. Eng. Chem. 2021, 102, 302-311. https://doi.org/10.1016/j.jiec.2021.07.016.
[25] Luo, Z.G.; Zhang, Y.; Wang, H.; Wan, S.; Song, L.F.; Liao, B.K.; Guo, X.P. Modified nano-lignin as a novel biomass-derived corrosion inhibitor for enhanced corrosion resistance of carbon steel. Corros. Sci. 2024, 227, 111705. https://doi.org/10.1016/j.corsci.2023.111705.
[26] Supardi, J.; Rizal, S.; Ali, N.; Fonna, S.; Ikramullah, I.; Arifin, A.K. Investigating an environmentally in aqueous solution of potacium chloride. J. Mater. Res. Technol. 2024, 29, 303-310. https://doi.org/10.1016/j.jmrt.2024.01.137.
[29] Wan, S.; Zhang, T.; Chen, H.; Liao, B.K.; Guo X.P. Kapok leaves extract and synergistic iodide as novel effective corrosion inhibitors for Q235 carbon steel in H2SO4 medium. Ind. Crop. Prod. 2022, 178, 114649. https://doi.org/10.1016/j.indcrop.2022.114649.
Comments 2: How does the combination of the three plant extracts with KI enhance the corrosion inhibition efficiency compared to their individual use?
Response 2: Thank you for pointing this out. We agree with this comment. With the combination of the three plant extracts with KI, parts of I- were changed to I3-. The I- and I3- owned more polarizable and adsorbable properties than SO42-, which were easier to adsorb on the Q235 surface. Furthermore, the I- and I3- enhanced the electronegativity of Q235 surface, thus facilitating the adsorption of the three positively charged plant extracts. It contributed to form a more uniform and denser protective layer, which further slowed the corrosion rate of Q235. Thus, the combination of the three plant extracts with KI enhance the corrosion inhibition efficiency compared to their individual use.
The more detailed explanation as follows, which was provided in section 3.8 from the line of 377 to 384 of the revised manuscript.
“After adding KI, firstly, the anions of I- and I3- exhibited stronger specificity compared to SO42-, leading to their increased adsorption on Q235 steel surface and subsequent augmentation of negative charge. Then, substantial adsorption of AAEHx+/CIEHx+/CMEHx+ onto the anionic layer formed by I- and I3- acting as a bridge. This resulted in the formation of a more comprehensive adsorption film layer that further amplified the corrosion inhibition effect. Additionally, the defects in the adsorption film could be filled by AAE/CIE/CME through chemisorption, resulting in a denser adsorption film and an enhanced corrosion inhibition effect.”
Comments 3: What is the significance of the observed synergistic effect between the plant extracts and KI in terms of the adsorption behavior on the Q235 steel surface?
Response 3: Thank you for pointing this out. We agree with this comment. The significance of the observed synergistic effect between the plant extracts and KI in terms of the adsorption behavior on the Q235 steel surface is to enhance the electronegativity of Q235 surface, thus facilitating the adsorption than the three plant extracts and KI individual. It contributed to form a more uniform and denser protective layer on Q235 surface, which further slowed the corrosion rate of Q235 in a long time.
Comments 4: How do the electrochemical impedance spectroscopy (EIS) results support the formation of a protective layer by the corrosion inhibitors and what information can be obtained from the equivalent circuits used for data fitting?
Response 4: Thank you for pointing this out. We agree with this comment. From the electrochemical impedance spectroscopy (EIS) results, the increase of the arc radius in Nyquist plot, the modulus value in Bode plot at 0.01 Hz, the peak value and the width range of phase angle under the presence of the corrosion inhibitors compared to those in the blank sample were used to support the formation of a protective layer by the corrosion inhibitors, which had been explained in the section 3.4 from the line of 204 to 212 of the revised manuscript.
The corresponding detailed explanation as follows:
“The Nyquist plots in Fig. 4 (a1), (c1), and (e1) exhibited capacitive reactance arcs with larger the high frequency regions and inductive reactance arcs with smaller low frequency regions. The capacitive reactance arcs were flatness, resulting from the non-uniformity of metal surface. As the concentration of three extracts increased, the radius of capacitive reactance arc and impedance mode at 0.01 Hz gradually increased, while the phase angle range became widened. However, there were no change in the shape of Nyquist plots, suggesting that a protective film layer formed on Q235 steel surface, hindering corrosion process without altering the electrochemical corrosion mechanism.”
The information can be obtained from the equivalent circuits used for data fitting as follows with detailed explanation, which had been explained in brief in the section 3.4 from the line of 230 to 232 of the revised manuscript.
(1) Rs, represented solution resistance, which was used to described the conductivity of an ion in a solution.
(2) CPEdl, represented double layer capacitance, which was closely related to the physical and chemical properties of the electrode surface.
(3) Rct, represented charge transfer resistance, which was mainly used to describe the resistance of charge transfer from the electrolyte to the electrode at the electrode/electrolyte interface.
(4) L, represented inductance, which was utilized to characterize the behavior of induced reactance in certain electrochemical systems, particularly in the high-frequency domain.
(5) RL, represented inductive resistance, which was a parameter that describes the resistance characteristics of the induction circuit in an electrochemical system.
Comments 5: What are the possible reasons for the long-term corrosion inhibition performance of the three compound inhibitors and how does it relate to the stability of the adsorption film?
Response 5:Thank you for pointing this out. We agree with this comment. The possible reasons for the long-term corrosion inhibition performance of the three compound inhibitors are two as follows. The first one is the adsorption effect. The corrosion inhibitor molecules were firmly adsorbed on the Q235 surface via physicochemical interactions by adsorption effect. The second one is the formation of protective film. With the adsorption effect, a uniform, dense, and stable adsorption protective film formed, which effectively prevented contact between the corrosive medium and the Q235 substrate, thereby reducing the corrosion rate of Q235. What’s more, the stability of the adsorption film could continue to prevent the contact between the corrosive medium and the Q235 substrate, thus the three compound inhibitors could further maintain long-term corrosion inhibition performance.
Comments 6: Based on the results of this study, can you propose other potential applications or areas of research for plant-extracted corrosion inhibitors from the Asteraceae family?
Response 6: Thank you for pointing this out. We agree with this comment. Based on the results of this study, other potential applications or areas of research for plant-extracted corrosion inhibitors from the Asteraceae family can be two primary points. Firstly, it involves utilizing other Compositae plants for the extraction and preparation of novel corrosion inhibitors. Secondly, it aims to predict the potential corrosion inhibition properties and assess the research value of unreported plant extracts within this family.
Reviewer 2 Report
Comments and Suggestions for Authors
The manuscript presents a thorough investigation into the corrosion inhibition properties of plant extracts from the Asteraceae family on Q235 steel in sulfuric acid. The study is well-structured, and the findings contribute valuable insights into eco-friendly corrosion inhibitors. However, several areas require clarification and improvement to enhance the overall quality of the paper.
1. The abstract effectively summarizes the study. It could benefit from a clearer statement of the research problem and objectives at the beginning.
2. Authors need to mention the type of electrochemical tests in the abstract.
3. Clarify the rationale behind selecting specific plant extracts from the Asteraceae family. What criteria were used for their selection?
4. The methodology is detailed, but it would be beneficial to include more information on the extraction process, such as specific conditions (temperature, time) and why these were chosen.
5. Considering the poor reproducibility of electrochemical measurements, deviations need to be given in Tables 1-3.
6. In Fig. 2, black line did not reach a stable state. How to determine the stable state?
7. It would strengthen your discussion to include comparisons with other recent studies on plant-based corrosion inhibitors. How do your findings align or differ from existing literature regarding Asteraceae family plants or other families? The relevant work can be added:
https://doi.org/10.1007/s13369-021-05699-0
https://doi.org/10.1007/s40735-021-00540-5
https://doi.org/10.3390/ma17215202
8. In the PDP results, authors noted that the anodic reaction inhibition was not significant compared to cathodic inhibition. It is needed to provide more insight into why these extracts predominantly inhibit cathodic reactions. How does this affect their overall effectiveness as corrosion inhibitors?
9. Ensure that all figures and tables are clearly labeled and referenced in the text.
Author Response
Comments 1: The abstract effectively summarizes the study. It could benefit from a clearer statement of the research problem and objectives at the beginning.
Response 1: Thank you for pointing this out. We agree with this comment. A clearer statement of the research problem and objectives has been added at the beginning of the Abstract part from the line of 14 to 17 of the revised manuscript.
The added details as follows:
“Nowadays, the development of plant extracts as corrosion inhibitors to protect metals from corrosion is a popular research direction. However, given the vast diversity of plant species in nature, it is imperative to explore effective methods to improve screening efficiency in order to quickly identify the corrosion inhibition potential of plants.”
Comments 2: Authors need to mention the type of electrochemical tests in the abstract.
Response 2: Thank you for pointing this out. We agree with this comment. The type of electrochemical tests in the abstract has been added with red font in Abstract part from the line of 23 to 25 of the revised manuscript.
The corresponding details with adding contents as follows:
“The corrosion inhibition behavior and corresponding mechanism were systematically investigated by some electrochemical tests (open circuit potential, potentiodynamic polarization and electrochemical impedance spectroscopy) and surface characterizations (Fourier transform infrared spectroscopy, Scanning Electron Microscopy and X-ray photoelectron spectroscopy).”
Comments 3: Clarify the rationale behind selecting specific plant extracts from the Asteraceae family. What criteria were used for their selection?
Response 3: Thank you for pointing this out. We agree with this comment. The rationale and criteria behind selecting specific plant extracts from the Asteraceae family are mainly derived from different genera of the selected plants. It is essential to verify that different genera of Compositae contain similar components with corrosion inhibition properties for achieving research objectives of this article. It contributes to enhance the reliability and applicability of using plant taxonomy to predict the corrosion inhibition potential of untested plant extracts and enhance screening efficiency.
Comments 4: The methodology is detailed, but it would be beneficial to include more information on the extraction process, such as specific conditions (temperature, time) and why these were chosen.
Response 4: Thank you for pointing this out. We agree with this comment. The reasons of choosing such as specific conditions (temperature, time) on the extraction process are as follows.
(1) Based on the previous research references of [58, 60] in abstract part, Soxhlet extraction method was selected in this article, which had been demonstrated to be effectively utilized for the efficient extraction of Compositae.
(2) There are both polar and non-polar groups in the three plant extracts in this paper. Polar groups can be well dissolved in water, while non-polar groups can be well dissolved in polar solvents such as methanol and ethanol. To efficiently extract the three plants of Compositae, a method of 1:1 ratio of water to ethanol was employed, which had been demonstrated to be effectively utilized in our research group's previous work [63, 66, 67].
(3) As for specific conditions (temperature, time), they were performed for 2h at 80 ℃ in condensing reflux step, which also had been demonstrated to be effectively utilized in our research group's previous work [63, 66, 67].
[58] Zhang, H.B.; Ni, Z.H.; Wu, H.T.; Xu, Z.D.; Zhang, W.Y.; Huang, H.X.; Zhou, Q.; Yue, X.G.; Bao, J.K.; Li, X.M. Corrosion Inhibition of Carbon Steel in Hydrochloric Acid by Chrysanthemum Indicum Extract. Int. J. Electrochem. Sc. 2020, 15, 5487-5499. https://doi.org/10.20964/2020.06.22.
[60] Kalkhambkar, A.G.; Rajappa, S.K.; Manjanna, J.; Malimath, G.H. Saussurea obvallatta leaves extract as a potential eco-friendly corrosion inhibitor for mild steel in 1 M HCl. Inorg. Chem. Commun. 2022, 143, 109799. https://doi.org/10.1016/j.inoche.2022.109799.
[63] Wang, Y.B.; Li, L.J.; He, J.B.; Sum, B.J. Extract of Silybum marianum (L.) Gaertn Leaves as a Novel Green Corrosion Inhibitor for Carbon Steel in Acidic Solution. Materials. 2024, 17. https://doi.org/10.3390/ma17194794.
[66] Alhaidar, B.; Reeshah, F.; Jammoal, Y. The Possibility of Using Barley Grains Extract as an Acidic Corrosion Inhibitor for Carbon Steel. Baghdad. Sci. J. 2024, 21, 1306-1317. https://doi.org/10.21123/bsj.2023.8501.
[67] Zhu, L.; Fan, J.; Huang, H.; Guo, L.; Zhu, M.; Zheng, X.; Obot, I. B. Inhibitive effect of different solvent fractions of bamboo shoots extract on the corrosion of mild steel in 0.5 mol/L H2SO4 solution. J. Mol. Struct. 2021, 1243. https://doi.org/10.1016/j.molstruc.2021.130852.
Comments 5: Considering the poor reproducibility of electrochemical measurements, deviations need to be given in Tables 1-3.
Response 5: Thank you for pointing this out. We agree with this comment. The deviations of electrochemical measurements have been added in Table 1 in section 3.3, Table 2 in section 3.4 and Table 3 in section 3.5 with red font of the revised manuscript.
The added details with deviations in Table 1, Table 2 and Table 3 as follows:
Table 1. Electrochemical parameters of Q235 steel in 0.5 mol/L H2SO4 with different extracts at various concentrations.
|
concentration (mg/L) |
Ecorr (V) |
Icorr (mA·cm-2) |
η (%) |
blank |
-- |
-0.46 |
1.21 ± 0.06 |
-- |
KI |
-- |
-0.47 |
0.32 ± 0.009 |
73.30 ± 0.15 |
AAE |
50 |
-0.45 |
0.782 ± 0.025 |
35.87 ± 0.59 |
100 |
-0.46 |
0.63 ± 0.032 |
47.77 ± 0.10 |
|
200 |
-0.46 |
0.67 ± 0.027 |
44.63 ± 0.25 |
|
AAE + 60 mg/L KI |
50 |
-0.45 |
0.12 ± 0.0054 |
90.25 ± 0.21 |
100 |
-0.46 |
0.08 ± 0.0013 |
93.69 ± 0.32 |
|
200 |
-0.45 |
0.04 ± 0.0025 |
96.69 ± 0.01 |
|
CIE |
50 |
-0.46 |
0.64 ± 0.03 |
47.02 ± 0.17 |
100 |
-0.47 |
0.44 ± 0.011 |
63.55 ± 0.05 |
|
200 |
-0.47 |
0.32± 0.0076 |
73.55 ± 0.09 |
|
CIE + 60 mg/L KI |
50 |
-0.46 |
0.15 ± 0.0016 |
87.93 ± 0.36 |
100 |
-0.45 |
0.07 ± 0.001 |
94.36 ± 0.16 |
|
200 |
-0.46 |
0.05 ± 0.0008 |
95.74 ± 0.11 |
|
CME |
50 |
-0.46 |
1.11 ± 0.0016 |
8.26 ± 0.23 |
100 |
-0.46 |
0.69 ± 0.002 |
43.22 ± 0.39 |
|
200 |
-0.46 |
0.49 ± 0.01 |
59.42 ± 0.06 |
|
CME + 60 mg/L KI |
50 |
-0.45 |
0.06 ± 0.0002 |
94.97 ± 0.06 |
100 |
-0.46 |
0.05 ± 0.0045 |
95.67 ± 0.17 |
|
200 |
-0.45 |
0.02 ± 0.006 |
98.26 ± 0.06 |
Table 2. Fitting results and corrosion inhibition efficiency of impedance spectra at different concentrations of different extracts.
|
concentration (mg/L) |
RS (Ω cm2) |
CPEdl (μF cm-2) |
n |
RL (Ω cm2) |
L (µH cm2) |
Rct (Ω cm2) |
ηEIS (%) |
blank |
-- |
2.53 ± 0.27 |
2.31×10-4 |
0.90 |
-- |
-- |
8.36 ± 0.43 |
-- |
KI |
-- |
0.60 ± 0.12 |
1.81×10-4 |
0.86 |
-- |
-- |
35.27 ± 6.84 |
76.30 ± 1.17 |
AAE |
50 |
1.98 ± 0.2 |
2.54×10-4 |
0.90 |
152.62 ± 13.54 |
319.91 |
15.30 ± 2.17 |
45.36 ± 1.53 |
100 |
0.85 ± 0.03 |
1.82×10-4 |
0.89 |
200.21 ± 10.81 |
257.16 |
20.16 ± 2.62 |
58.53 ± 0.99 |
|
200 |
2.06 ± 0.22 |
1.75×10-4 |
0.90 |
184.86 ± 25.82 |
400.03 |
24.42 ± 2.89 |
65.77 ± 0.70 |
|
AAE + KI 60 mg/L |
50 |
1.82 ± 0.14 |
1.68×10-4 |
0.84 |
-- |
-- |
77.84 ± 8.92 |
89.26 ± 0.21 |
100 |
1.48 ± 0.08 |
1.19×10-4 |
0.86 |
-- |
-- |
171.53 ± 30.18 |
95.13 ± 0.20 |
|
200 |
1.37 ± 0.15 |
9.14×10-5 |
0.86 |
-- |
-- |
225.60 ± 48.34 |
96.29 ± 0.22 |
|
CIE |
50 |
1.85 ± 0.26 |
1.75×10-4 |
0.91 |
171.45 ± 19.25 |
490.80 |
20.96 ± 2.87 |
60.11 ±1.04 |
100 |
1.63 ± 0.07 |
1.54×10-4 |
0.90 |
295.80 ± 35.11 |
2192.00 |
30.08 ± 5.68 |
72.21 ± 1.31 |
|
200 |
1.60 ± 0.33 |
1.33×10-4 |
0.90 |
133.96 ± 21.08 |
1419.36 |
34.06 ± 6.33 |
75.46 ± 1.13 |
|
CIE + KI 60 mg/L |
50 |
2.02 ± 0.11 |
1.31×10-4 |
0.85 |
-- |
-- |
88.93 ± 12.09 |
90.60 ± 0.24 |
100 |
1.58 ± 0.18 |
1.00×10-4 |
0.86 |
-- |
-- |
182.46 ± 34.23 |
95.41 ± 0.20 |
|
200 |
1.51 ± 0.13 |
9.97×10-5 |
0.88 |
-- |
-- |
239.50 ± 63.71 |
96.50 ± 0.31 |
|
CME |
50 |
1.86 ± 0.41 |
1.93×10-4 |
0.91 |
144.62 ± 10.12 |
399.31 |
17.46 ± 2.26 |
52.12 ± 1.14 |
100 |
1.62 ± 0.24 |
1.87×10-4 |
0.90 |
184.34 ± 19.8 |
588.83 |
19.14 ± 2.88 |
56.32 ± 1.36 |
|
200 |
1.83 ± 0.31 |
1.63×10-4 |
0.89 |
252.10 ± 26.4 |
885.06 |
27.82 ± 4.11 |
69.95 ± 0.90 |
|
CME + KI 60 mg/L |
50 |
1.83 ± 0.16 |
1.13×10-4 |
0.86 |
-- |
-- |
140.72 ± 29.63 |
94.06 ± 0.34 |
100 |
1.25 ± 0.06 |
8.87×10-5 |
0.87 |
-- |
-- |
260.44 ± 45.15 |
96.79 ± 0.13 |
|
200 |
1.54 ± 0.13 |
7.80×10-5 |
0.87 |
-- |
-- |
336.54 ± 58.48 |
97.52 ± 0.10 |
Table 3. Fitting results of EIS curves of Q235 steel soaking time in 0.5 mol/L H2SO4 at 200 mg/L combined with 60 mg/L KI at different immersion times.
|
Time (h) |
Rs (Ω cm2) |
CPEdl (µF cm-2) |
n |
Rct (Ω cm2) |
ηEIS (%) |
blank |
-- |
2.53 ± 0.27 |
2.31×10-4 |
0.90 |
8.36 ± 0.43 |
-- |
KI |
-- |
0.60 ± 0.12 |
1.81×10-4 |
0.86 |
35.27 ± 6.84 |
76.30 ± 1.17 |
AAE 200 mg/L + KI 60 mg/L |
1 |
1.25 ± 0.11 |
6.64×10-5 |
0.88 |
370.83 ± 58.54 |
97.75 ± 0.08 |
2 |
1.25 ± 0.15 |
5.03×10-5 |
0.90 |
536.12 ± 84.98 |
98.44 ± 0.05 |
|
4 |
1.32 ± 0.09 |
4.36×10-5 |
0.90 |
613.14 ± 91.51 |
98.64 ± 0.05 |
|
8 |
1.31 ± 0.13 |
4.63×10-5 |
0.91 |
557.76 ± 63.40 |
98.50 ± 0.03 |
|
16 |
1.03 ± 0.04 |
8.63×10-5 |
0.82 |
273.33 ± 40.26 |
96.94 ± 0.09 |
|
24 |
0.92 ± 0.05 |
1.19×10-4 |
0.69 |
169.71 ± 20.04 |
95.07 ± 0.10 |
|
48 |
1.70 ± 0.21 |
1.44×10-4 |
0.64 |
116.90 ± 10.80 |
92.85 ± 0.10 |
|
CIE 200 mg/L + KI 60 mg/L
|
1 |
1.83 ± 0.26 |
1.05×10-4 |
0.88 |
180.26 ± 26.88 |
95.36 ± 0.14 |
2 |
1.87 ± 0.20 |
6.92×10-5 |
0.89 |
299.93 ± 58.92 |
97.21 ± 0.14 |
|
4 |
1.87 ± 0.17 |
6.75×10-5 |
0.89 |
355.70 ± 65.69 |
97.65 ± 0.11 |
|
8 |
1.86 ± 0.24 |
6.88×10-5 |
0.88 |
340.39 ± 79.61 |
97.54 ± 0.17 |
|
16 |
1.80 ± 0.20 |
7.84×10-5 |
0.82 |
297.71 ± 60.08 |
97.19 ± 0.15 |
|
24 |
1.79 ± 0.18 |
1.10×10-4 |
0.81 |
236.50 ± 52.80 |
96.47 ± 0.23 |
|
48 |
0.57 ± 0.05 |
1.62×10-4 |
0.84 |
80.27 ± 16.26 |
89.59 ± 0.56 |
|
|
1 |
1.89 ± 0.22 |
6.32×10-5 |
0.89 |
535.76 ± 101.03 |
98.44 ± 0.07 |
|
2 |
1.85 ± 0.24 |
3.97×10-5 |
0.90 |
796.04 ± 104.89 |
98.95 ± 0.03 |
|
4 |
1.84 ± 0.13 |
3.27×10-5 |
0.90 |
893.13 ± 113.25 |
99.06 ± 0.02 |
CME 200 mg/L + KI 60 mg/L |
8 |
1.85 ± 0.17 |
4.11×10-5 |
0.89 |
739.92 ± 108.40 |
98.87 ± 0.03 |
|
16 |
1.56 ± 0.19 |
6.62×10-5 |
0.84 |
444.41 ± 80.25 |
98.12 ± 0.08 |
|
24 |
1.28 ± 0.10 |
1.12×10-4 |
0.79 |
223.46 ± 42.26 |
96.26 ± 0.18 |
|
48 |
0.59 ± 0.05 |
1.24×10-4 |
0.75 |
120.60 ± 15.84 |
93.07 ± 0.17 |
Comments 6: In Fig. 2, black line did not reach a stable state. How to determine the stable state?
Response 6: Thank you for pointing this out. We agree with this comment. Based on the previous published researches of [27] in Introduction part and [71] in section 3.2 of the article, OCP can be considered stable if the OCP changes less than 10 millivolts in one minute or the OCP oscillates less than dE/dt< 5×10-4 throughout the test. The changes of OCP after the 1800s meet the above conditions, thus black line can be considered to reach a stable state.
[27] Zhou, Z.Y.; Min, X.H.; Wan, S.; Liu, J.H.; Liao, B.K.; Guo, X.P. A novel green corrosion inhibitor extracted from waste feverfew root for carbon steel in H2SO4 solution. Results. in. Engineering. 2023, 17, 100971. https://doi.org/10.1016/j.rineng.2023.100971.
[71] Chen, Q.H.; Zhou, Z.Y.; Feng, M.T.;, He, J.H.; Xu, Y.Q.; Liao, B.K. Unraveling the inhibitive performance and adsorption behavior of expired compound glycyrrhizin tablets as an eco-friendly corrosion inhibitor for copper in acidic medium. J. Taiwan. Inst. Chem. E. 2025, 168, 105913. https://doi.org/10.1016/j.jtice.2024.105913.
Comments 7: It would strengthen your discussion to include comparisons with other recent studies on plant-based corrosion inhibitors. How do your findings align or differ from existing literature regarding Asteraceae family plants or other families? The relevant work can be added:
https://doi.org/10.1007/s13369-021-05699-0
https://doi.org/10.1007/s40735-021-00540-5
https://doi.org/10.3390/ma17215202
Response 7: Thank you for pointing this out. We agree with this comment. The three articles you provided really strengthen our discussion of this paper, which have been added with the number of [15], [16], [17] in Introduction part and other parts of the revised manuscript.
In addition, our findings in this article are align with existing literature among [57-65] in Introduction part regarding Asteraceae family plants. Both of them have similar active components with groups and composition of corrosion products with bonding mechanism. The relevant work has been added in section 3.1 from the line of 145 to 146, 149 to 150 and section 3.6 from the line of 318 to 319 of the revised manuscript.
The added details as follows:
“The majority of the structures exhibited similarities to those found in other members of the Asteraceae family [57, 58, 60, 62].” (line of 145 to 146)
“The main components also found in other Asteraceae family plants [61].” (line of 149 to 150)
“The composition of corrosion products and the bonding mechanism can also be observed in other plants of Asteraceae family [63].” (line of 318 to 319)
[15] Kaur, J.; Daksh, N.; Saxena, A. Corrosion Inhibition Applications of Natural and Eco-Friendly Corrosion Inhibitors on Steel in the Acidic Environment: An Overview. Arab. J. Sci. Eng. 2022, 47, 57-74. https://doi.org/10.1007/s13369-021-05699-0.
[16] Panchal, J.; Shah, D.; Patel, R.; Shah, S.; Prajapati, M.; Shah, M. Comprehensive Review and Critical Data Analysis on Corrosion and Emphasizing on Green Eco-friendly Corrosion Inhibitors for Oil and Gas Industries Journal. of. Bio-. and. Tribo-Corrosion. 2021, 107. https://doi.org/10.1007/s40735-021-00540-5.
[17] Faraji, M.; Pezzato, L.; Yazdanpanah, A.; Nardi, G.; Esmailzadeh, M.; Calliari, I. Effect of Natural Inhibitors on the Corrosion Properties of Grade 2 Titanium Alloy. Materials. 2024, 17. https://doi.org/10.3390/ma17215202.
[57] Wang, Q.H.; Liu, L.; Zhang, Q.; Wu, X.D.; Zheng, H.H.; Gao, P.; Zeng, G.M.; Yan, Z.T.; Sun, Y.; Li, X.M. Insight into the anti-corrosion performance of Artemisia argyi leaves extract as eco-friendly corrosion inhibitor for carbon steel in HCl medium. Sustain. Chem. Pharm. 2022, 27, 100710. https://doi.org/10.1016/j.scp.2022.100710.
[58] Zhang, H.B.; Ni, Z.H.; Wu, H.T.; Xu, Z.D.; Zhang, W.Y.; Huang, H.X.; Zhou, Q.; Yue, X.G.; Bao, J.K.; Li, X.M. Corrosion Inhibition of Carbon Steel in Hydrochloric Acid by Chrysanthemum Indicum Extract. Int. J. Electrochem. Sc. 2020, 15, 5487-5499. https://doi.org/10.20964/2020.06.22.
[59] Mourya, P.; Banerjee, S.; Singh, M.M. Corrosion inhibition of mild steel in acidic solution by Tagetes erecta (Marigold flower) extract as a green inhibitor. Corros. Sci. 2014, 85, 352-363. http://dx.doi.org/10.1016/j.corsci.2014.04.036.
[60] Kalkhambkar, A.G.; Rajappa, S.K.; Manjanna, J.; Malimath, G.H. Saussurea obvallatta leaves extract as a potential eco-friendly corrosion inhibitor for mild steel in 1 M HCl. Inorg. Chem. Commun. 2022, 143, 109799. https://doi.org/10.1016/j.inoche.2022.109799.
[61] Hassannejad, H.; Nouri, A. Sunflower seed hull extract as a novel green corrosion inhibitor for mild steel in HCl solution. J. Mol. Liq. 2018, 254, 377-382. https://doi.org/10.1016/j.molliq.2018.01.142.
[62] Kouache, A.; Khelifa, A.; Boutoumi, H.; Moulay, S.; Feghoul, A.; Idir, B.; Aoudj, S. Experimental and theoretical studies of Inula viscosa extract as a novel eco-friendly corrosion inhibitor for carbon steel in 1 M HCl. J. Adhes. Sci. Technol. 2022, 36, 988-1016. https://doi.org/10.1080/01694243.2021.1956215.
[63] Wang, Y.B.; Li, L.J.; He, J.B.; Sum, B.J. Extract of Silybum marianum (L.) Gaertn Leaves as a Novel Green Corrosion Inhibitor for Carbon Steel in Acidic Solution. Materials. 2024, 17. https://doi.org/10.3390/ma17194794.
[64] Beniaich, G.; Beniken, M.; Salim, R.; Arrousse, N.; Ech-chibi, E.; Rais, Z.; Sadiq, A.; Nafidi, H.A.; Bin Jardan, Y.A.; Bourhia, M.; Taleb, M. Anticorrosive Effects of Essential Oils Obtained from White Wormwood and Arar Plants. Separations. 2023, 10. https://doi.org/10.3390/separations10070396.
[65] Cang, H.; Shi, W.Y.; Shao, J.L.; Xu, Q. Study of Stevia rebaudiana Leaves as Green Corrosion Inhibitor for Mild Steel in Sulphuric Acid by Electrochemical Techniques. Int. J. Electrochem. Sc. 2012, 7, 3726-3736. https://doi.org/WOS000304413000080.
Comments 8: In the PDP results, authors noted that the anodic reaction inhibition was not significant compared to cathodic inhibition. It is needed to provide more insight into why these extracts predominantly inhibit cathodic reactions. How does this affect their overall effectiveness as corrosion inhibitors?
Response 8: Thank you for pointing this out. We agree with this comment. The more insight into why these extracts predominantly inhibit cathodic reactions are as follows. Based on the previous research references of [83-87] in section of 3.8, in acidic medium, these three plant extracts of AAE/CIE/CME will undergo the protonation reaction with H+ to generate the protonated substance of AAEHx+/CIEHx+/CMEHx+, as the Eqs. (3~5):
AAE + xH+ → AAEHx+ (3)
CIE + xH+ → CIEHx+ (4)
CME + xH+ → CMEHx+ (5)
The protonation reaction can significantly reduce the reduction rate of H+, thereby decelerating the cathodic reaction. Furthermore, the corrosion inhibitor molecules were firmly adsorbed on the Q235 surface via physicochemical interactions by adsorption effect. With the adsorption effect, a uniform, dense, and stable adsorption protective film formed and covered on the Q235 surface. Consequently, AAE/CIE/CME or AAE/CIE/CME/KI is a mixed-type inhibitor based on a geometric coverage effect electrochemical inhibition mechanism. Therefore, these extracts predominantly inhibit cathodic reactions. The more detailed insights have been provided in section 3.8 of the revised manuscript.
Under this promotion of these plant extracts of AAE/CIE/CME or AAE/CIE/CME/KI predominantly inhibit cathodic reactions, the electrochemical corrosion reactions of Q235 are significantly decelerated, thereby enhancing the overall effectiveness of these corrosion inhibitors.
[83] Wang, H.; Deng, S.; Du, G.; Li, X. Synergistic mixture of Eupatorium adenophora spreng leaves extract and KI as a novel green inhibitor for steel corrosion in 5.0 M H3PO4. J. Mater. Res. Technol. 2023, 23, 5082-5104. https://doi.org/10.1016/j.jmrt.2023.02.160.
[84] Teng, X.Y.; Deng, S.D.; Li, X.H. Synergistic corrosion inhibition of rubber seed extract with KI on cold rolled steel in sulfuric acid solution. J. Taiwan. Inst. Chem. E. 2024, 161, 105564. https://doi.org/10.1016/j.jtice.2024.105564.
[85] Du, P.F.; Yang, H.F.; Deng, S.D.; Li, X.H. Synergistic inhibition of Mikania micrantha extract with iodide ion on the corrosion of cold rolled steel in trichloroacetic acid medium. J. Ind. Eng. Chem. 2024, 139, 358-377. https://doi.org/10.1016/j.jiec.2024.05.014.
[86] Wang, Y.; Qiang, Y.J.; Zhi, H.; Ran, B.Y.; Zhang, D.W. Evaluating the synergistic effect of maple leaves extract and iodide ions on corrosion inhibition of Q235 steel in H2SO4 solution. J. Ind. Eng. Chem. 2023, 117, 422-433. https://doi.org/10.1016/j.jiec.2022.10.030.
[87] Qiu, L.; Li, X.H.; Xu, D.K.; Shao, D.D.; Du, G.B.; Deng, S.D. Comparison of the corrosion inhibition property on cold rolled steel in sulfuric acid media between reflux and ultrasound extracts from rapeseed meal. Ind. Crop. Prod. 2024, 216, 118809. https://doi.org/10.1016/j.indcrop.2024.118809.
Comments 9: Ensure that all figures and tables are clearly labeled and referenced in the text.
Response 9: Thank you for pointing this out. We agree with this comment. All figures and tables have been clearly labeled and referenced in the text. Several original images requiring modification have been marked with a blue background and strikethrough, followed immediately by the corresponding revised images as in the revised manuscript.